# Adaptive Diffusion in Graph Neural Networks

**Jialin Zhao**
Tsinghua University
zjl19970607@gmail.com

**Yuxiao Dong**[*]
Microsoft Research
ericdongyx@gmail.com

**Ming Ding**
Tsinghua University
dm18@mails.tsinghua.edu.cn

**Evgeny Kharlamov**
Bosch Center for Artificial Intelligence
Evgeny.Kharlamov@de.bosch.com

**Jie Tang**[†]
Tsinghua University
jietang@tsinghua.edu.cn

## Abstract

The success of graph neural networks (GNNs) largely relies on the process of aggregating information from neighbors defined by the input graph structures. Notably, message passing based GNNs, e.g., graph convolutional networks, leverage the immediate neighbors of each node during the aggregation process, and recently, graph diffusion convolution (GDC) is proposed to expand the propagation neighborhood by leveraging generalized graph diffusion. However, the neighborhood size in GDC is manually tuned for each graph by conducting grid search over the validation set, making its generalization practically limited. To address this issue, we propose the adaptive diffusion convolution (ADC)[*] strategy to automatically learn the optimal neighborhood size from the data. Furthermore, we break the conventional assumption that all GNN layers and feature channels (dimensions) should use the same neighborhood size for propagation. We design strategies to enable ADC to learn a dedicated propagation neighborhood for each GNN layer and each feature channel, making the GNN architecture fully coupled with graph structures—the unique property that differs GNNs from traditional neural networks. By directly plugging ADC into existing GNNs, we observe consistent and significant outperformance over both GDC and their vanilla versions across various datasets, demonstrating the improved model capacity brought by automatically learning unique neighborhood size per layer and per channel in GNNs.

## 1 Introduction

Graph neural networks (GNNs) are a type of neural networks that can be directly coupled with graph-structured data [30, 41]. Specifically, graph convolution networks [12, 19] (GCNs) generalize the convolution operation to local graph structures, offering attractive performance for various graph mining tasks [15, 32, 37]. The graph convolution operation is designed to aggregate information from immediate neighboring nodes into the central node, which is also referred to as message passing [14]. To propagate information between nodes that are further away, multiple neural layers can be stacked

---

[*]Now at Meta AI and work done when at Microsoft.

[†]Jie Tang is the corresponding author.

[*]Code is available at https://github.com/abcbdf/ADC

35th Conference on Neural Information Processing Systems (NeurIPS 2021).

to go beyond the immediate hop of neighbors. To directly collect high-order information, spectral based GNNs leverage graph spectral properties to collect signals from global neighbors [6, 12, 17].

Though generating promising results, both strategies are limited to a pre-determined and fixed neighborhood for passing and receiving messages. Essentially, these methods have an implicit assumption that all graph datasets share the same size of receptive field during the message passing process. To break this, graph diffusion convolution (GDC) [21] was recently proposed to extend the discrete message passing process in GCN to a diffusion process, enabling it to aggregate information from a larger neighborhood. For each input graph, GDC hand-tunes the best neighborhood size for feature aggregation by grid-searching the parameters on the validation set, making its practical application limited and sensitive.

To eliminate the manual search process of the optimal propagation neighborhood in GDC, we propose the adaptive diffusion convolution (ADC) strategy that supports learning the optimal neighborhood from the data automatically. ADC achieves this by formalizing the task as a bilevel optimization problem [11], enabling the customized learning of *one* optimal propagation neighborhood size for each dataset. In other words, all GNN layers and feature channels (dimensions) share the same neighborhood size during message passing on each graph.

To further this direction, we also enable ADC to automatically learn a customized neighborhood size for each GNN layer and each feature channel from data. By learning a unique propagation neighborhood for each layer, ADC can empower GNNs to capture neighbors' information from diverse graph structures, which is fully dependent on the data and downstream learning objective. Similarly, by learning distinct neighborhood size for each feature channel, GNNs are then capable of selectively modeling each neighbor's multiple feature signals. Altogether, ADC makes GNNs fully coupled with the graph structures and all feature channels.

By design, ADC is a general plugin that can be directly applied to existing GNN models. By plugging it on several GNNs, we show that the upgraded GNNs can offer significant performance advances over their vanilla versions across a wide range of datasets. Furthermore, experimental results also show that by learning the propagation neighborhood size automatically, ADC can consistently outperform GDC, which customizes this for each dataset by grid search. Finally, we demonstrate that GNNs' model capacity can benefit from the better coupling between the its architecture, graph structures, and feature channels, that is, by learning dedicated neighborhood size for each GNN layer and feature channel.

## 2 Neighborhood Radius in GNNs

We focus on the problem of semi-supervised node classification. The input includes an undirected network $G = (V, E)$, where the node set $V$ contains of $n$ nodes $\{v_1, ..., v_n\}$ and $E$ is the edge set, and $\mathbf{A} \in R^{n \times n}$ is the symmetric adjacency matrix of graph $G$. Given the input feature matrix $\mathbf{X}$ and a subset of node label $\mathbf{Y}$, the task is to predict the labels of remaining nodes.

### 2.1 Neighborhood Radius in Message Passing Networks

The convolution operation on graphs can be described as the process of neighborhood feature aggregation or message passing [14]. The message passing graph convolutional networks can be simply defined as:

$$\mathbf{H}^{(l)} = \gamma^{(l)}(\varphi^{(l)}(\mathbf{H}^{(l-1)}), G) \tag{1}$$

where $\mathbf{H}^{(l)}$ denotes the hidden feature of layer $l$ with $\mathbf{H}^{(0)} = \mathbf{X}$ and $\mathbf{X}$ as the input feature, $\varphi(\cdot)$ denotes feature transformation and $\gamma(\cdot)$ denotes feature propagation. Take GCN [19] for example. The feature transformation and feature propagation functions correspond to $\varphi(\mathbf{H}) = \mathbf{H}W$, $\gamma(\hat{\mathbf{H}}, G) = \tilde{\mathbf{D}}^{-\frac{1}{2}}\tilde{\mathbf{A}}\tilde{\mathbf{D}}^{-\frac{1}{2}}\hat{\mathbf{H}}$, respectively, in which $\mathbf{D}$ is the diagonal degree matrix with $\tilde{\mathbf{D}}_{ii} = \sum_j \tilde{\mathbf{A}}_{ij}$, and $\hat{\mathbf{H}}$ denotes hidden feature after transformation. Note that GCN uses the adjacency matrix $\mathbf{A}$ with self loop, so it actually uses $\tilde{\mathbf{A}} = \mathbf{I} + \mathbf{A}$. To simplify the notations, we use $\mathbf{T}$ to denote $\tilde{\mathbf{D}}^{-\frac{1}{2}}\tilde{\mathbf{A}}\tilde{\mathbf{D}}^{-\frac{1}{2}}$.

Straightforwardly, the feature transformation function $\varphi(\cdot)$ describes how features transform inside each node and the feature propagation function $\gamma(\cdot)$ describes how features propagate between nodes. Essentially, how good a GNN model can utilize graph structures heavily depends on the design of the feature propagation function.

**Neighborhood radius $r$.** Most graph-based models can be represented as $\gamma^{(l)}(\hat{\mathbf{H}}, G) = f(\mathbf{T})\hat{\mathbf{H}}$, where $f(\mathbf{T})$ is a matrix that can be generated by $\mathbf{T}$. So $f(\mathbf{T})$ can be represented as $f(\mathbf{T}) = \sum_{k=0}^{\infty} \theta_k \mathbf{T}^k$. To quantify how far each node could aggregate features from, we define the neighborhood radius of a node as $r$:

$$r = \frac{\sum_{k=0}^{\infty} \theta_k k}{\sum_{k=0}^{\infty} \theta_k} \tag{2}$$

Here, $\theta_k$ denotes the influence from $k$-step-away nodes. For a large $r$, this means the model puts more emphasis on long distance nodes, i.e., global information. For a small $r$, this means the model amplifies local information.

**Neighborhood radius $r$ in GCN.** For GCN, the neighborhood radius $r = 1$, which is just range of nodes directly connected to it. To collect information beyond direct connections, it is required to stack multiple GCN layers to reach high-order neighborhoods.

**Neighborhood radius $r$ in multi-hop models.** There are attempts to improve GCN's feature propagation function from first-hop neighborhood to multi-hop neighborhood, such as MixHop [2], JKNet [38], and SGC [35]. For example, SGC [35] uses feature propagation function $\gamma(\hat{\mathbf{H}}, G) = \mathbf{T}^K \hat{\mathbf{H}}$, where $\mathbf{T} = \tilde{\mathbf{D}}^{-\frac{1}{2}} \tilde{\mathbf{A}} \tilde{\mathbf{D}}^{-\frac{1}{2}}$. In other words, the neighborhood radius $r = K$ for SGC, which is the range of neighborhoods to collect information from each GNN layer. However, for all multi-hop models, the discrete nature of hop numbers makes $r$ non-differentiable.

## 2.2 Neighborhood Radius in Graph Diffusion Convolution

Recently, a line of work has been focused on generalizing feature propagation from discrete hops to continuous graph diffusion [21, 36, 40]. Notably, graph diffusion convolution (GDC) addresses this by the following propagation setup [21]:

$$\gamma^{(l)}(\hat{\mathbf{H}}, G) = \sum_{k=0}^{\infty} \theta_k \mathbf{T}^k \hat{\mathbf{H}} \tag{3}$$

where $k$ is summed from 0 to infinity, making each node aggregate information from the whole graph.

In Eq.3, the weight coefficients should satisfy $\sum_{k=0}^{\infty} \theta_k = 1$ such that the signal strength is not amplified nor reduced through the propagation. The two commonly-used sets of weight coefficients [21, 36, 40] are generated from personalized PageRank ($\theta_k = \alpha(1 - \alpha)^k$) [27] and the heat kernel ($\theta_k = e^{-t} \frac{t^k}{k!}$) [22], respectively. In this work, we focus on heat kernel.

**Heat kernel.** Heat kernel incorporates prior knowledge into the GNN model, which means the feature propagation between nodes follows Newton's law of cooling [34], i.e., the feature propagation speed between two nodes is proportional to the difference between their features. Formally, this prior knowledge can be described as:

$$\frac{d\mathbf{x}_i(t)}{dt} = -\sum_{j \in N(i)} \tilde{\mathbf{A}}_{ij}(\mathbf{x}_i(t) - \mathbf{x}_j(t)) \tag{4}$$

where $N(i)$ denotes the neighborhood of node $i$, $\mathbf{x}_i(t)$ represents the feature of node $i$ after diffusion time $t$. This differential equation can be solved as:

$$\mathbf{X}(t) = \mathbf{H}_t \mathbf{X}(0) \tag{5}$$

where $\mathbf{X}(t)$ means the feature matrix after diffusion time $t$ and $\mathbf{H}_t = e^{-(\mathbf{I} - \mathbf{T})t}$ is the heat kernel.

**Neighborhood radius $r_h$ in diffusion models.** According to the definition of neighborhood radius Eq. 2, the heat kernel version of the GDC has neighborhood radius $r_h$ as

$$r_h = \frac{\sum_{k=0}^{\infty} \theta_k k}{\sum_{k=0}^{\infty} \theta_k} = \frac{\sum_{k=0}^{\infty} e^{-t} \frac{t^k}{k!} k}{\sum_{k=0}^{\infty} e^{-t} \frac{t^k}{k!}} = \frac{e^{-t} \sum_{k=0}^{\infty} \frac{t^k}{k!} k}{e^{-t} \sum_{k=0}^{\infty} \frac{t^k}{k!}} = \frac{e^{-t}(e^t t)}{e^{-t} e^t} = t \tag{6}$$

This suggests that $t$ is the neighborhood radius for the heat kernel based GDC, that is, $t$ becomes a perfect continuous substitute for the hop number in multi-hop models.

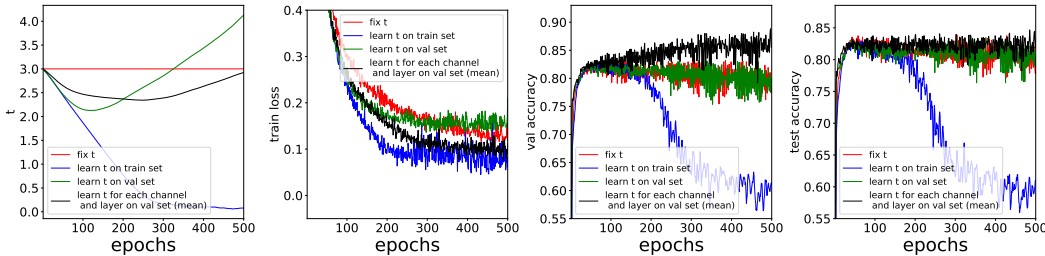

Figure 1: Comparison among fixing $t$ (red), training $t$ on train set (blue), training $t$ on validation set (green), and training $t$ separately for each feature channel on validation set (black). The results are reported by using the Heat Kernel version of GCN on the Cora dataset. Training $t$ on validation set can prevent overfitting.

## 3 Adaptive Diffusion Convolution

Recall that the heat kernel version of graph diffusion convolution (GDC) has the following feature propagation function as

$$\gamma^{(l)}(\hat{\mathbf{H}}, G) = e^{-\mathbf{L}t}\hat{\mathbf{H}} = \sum_{k=0}^{\infty} e^{-t}\frac{t^k}{k!}\mathbf{T}^k\hat{\mathbf{H}} \tag{7}$$

where the Laplacian matrix $\mathbf{L} = \mathbf{I} - \mathbf{T}$. For each graph dataset, it requires the manual grid search step to determine the neighborhood radius related parameter $t$. Moreover, $t$ is fixed for all feature channels and propagation layers in each dataset. In this work, we explore how to adaptively learn the neighborhood radius from data for each graph and further examine the potential to generalize it for different feature channels and GNN layers.

### 3.1 Training Neighborhood Radius

Diffusion convolution enables us to replace GNNs' discrete feature propagation function with the continuous heat kernel. Instead of hand-tuning $t$, we can calculate the gradient of $t$ and update $t$ to converge to an optimal neighborhood, which is the same to learning other weight and bias parameters in the model.

Figure 1 shows the training process of learning $t$. With more epochs, both $t$ and training loss decrease when learning on the training set (blue). Meanwhile, the validation and test accuracies drop dramatically as $t$ tends to zero (more epochs)—representing each node could only use its own features to predict the label. That is, learning $t$ directly on the training set causes overfitting. This phenomenon has also been observed in GDC [21]. The authors found that strong regularization on the difference of $\theta_{k+1} - \theta_k$ could help overcome the overfitting issue. However, that would require hand-tuning the regularization factor for every dataset, which is similar to hand-tuning $t$ itself, e.g., by grid search, further limiting the generalization of the model.

To address this issue, we propose a method of training $t$ by using the gradient of the model on the validation set. The goal for the model is to find $t^*$ that minimizes the validation loss $\mathcal{L}_{val}(t, w^*)$, where $w$ denotes all the other trainable parameters in the feature transformation function and $w^*$ denotes the set of parameters that minimize the training loss $\mathcal{L}_{train}(t, w)$. This strategy can be formalized as a bilevel optimization problem [3, 11]:

$$t^* = \arg\min_t \mathcal{L}_{val}(t, w^*(t)) \tag{8}$$

$$w^*(t) = \arg\min_w \mathcal{L}_{train}(t, w) \tag{9}$$

By doing so, every time we update $t$, we need to make $w$ converge to the optimal value, which is too expensive to train. An approximation method is to update $t$ every time we update $w$, that is,

$$w^{(e+1)} = w^{(e)} - \alpha_1 \bigtriangledown_w \mathcal{L}_{train}(t^{(e)}, w^{(e)}) \tag{10}$$

$$t^{(e+1)} = t^{(e)} - \alpha_2 \nabla_t \mathcal{L}_{val}(t^{(e)}, w^{(e+1)}) \tag{11}$$

where $e$ denotes the number of training epochs, $\alpha_1$ and $\alpha_2$ denote the learning rate on the training and validation sets, respectively. The similar idea has been proposed in the gradient-based hyperparameter tuning [24] and neural architecture search [23]. Figure 1 shows that using this method (green lines) helps avoid overfitting and thus offers better generalization, as there is no sign of test accuracy drop. Meanwhile, $t$ does not diminish to zero, indicating the meaningful learning of this parameter—neighborhood radius.

## 3.2  Training Neighborhood Radius for Each Layer and Channel

Conventional GNNs use the predetermined neighborhood radius for feature propagation. GDC proposes to use different neighborhood radius $t$ for different datasets by hand-tuning the values. The above method furthers this direction by automatically learning the radius $t$ from the given graph. This implies that one $t$ for one dataset, that is, the same $t$ for all GNN layers and all feature channels (dimensions).

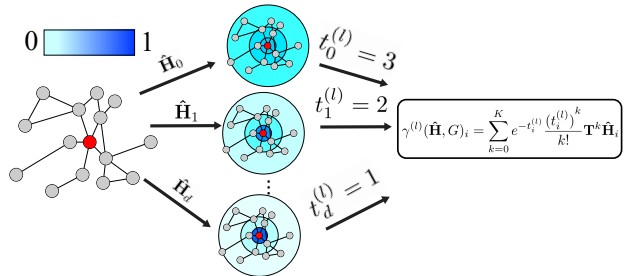

Figure 2: Illustration of the Adaptive Diffusion Convolution (ADC). For the hidden feature $\hat{\mathbf{H}}_i$ of feature channel $i$ in layer $l$, we train a separate feature propagation function $\gamma^{(l)}(\hat{\mathbf{H}}, G)_i$ with a unique neighborhood radius $t_i^{(l)}$. When $t$ is large (e.g., $t=3$), the contributions from close (e.g., in 1-hop) and distant neighbors (e.g., in 3-hop) have little difference (shown as the relatively similar color shading across different hops). When $t$ is small (e.g., $t=1$), the contributions from close neighbors are much more significant than from distant neighbors (shown as dark color concentrated around center).

**Adaptive diffusion convolution (ADC).** The natural question arises here is whether we can have a unique $t$ for each layer and channel, making them adaptive for the final learning objective. The obstacle that prevents previous models from achieving this lies in the infeasible challenge of hand-tuning or grid-searching the propagation function separately for each feature channel and GNN layer, given that as the number of parameters increases, the time complexity increases exponentially. However, the aforementioned strategy for updating $t$ during the training of the model empowers us to adaptively learn specific $t$ for all layers and all feature channels.

Straightforwardly, we have the adaptive diffusion convolution (ADC) by extending the feature propagation function in Eq. 7 for each layer and channel, that is, from $t$ to $t_i^{(l)}$,

$$\gamma^{(l)}(\hat{\mathbf{H}}, G)_i = \sum_{k=0}^{\infty} e^{-t_i^{(l)}} \frac{\left(t_i^{(l)}\right)^k}{k!} \mathbf{T}^k \hat{\mathbf{H}}_i \tag{12}$$

where $t_i^{(l)}$ denotes the neighborhood radius $t$ for the $l$-th layer and $i$-th channel, $\hat{\mathbf{H}}_i$ represents the $i$-th column of the hidden feature $\hat{\mathbf{H}}$, i.e., the feature on channel $i$, and $\gamma_i^{(l)}$ denotes the feature propagation function on the $l$-th layer and $i$-th channel. This feature propagation function enables the GNN to train a separate $t$ for each feature channel and layer, which is illustrative in Figure 2. In addition, Figure 1 (black lines) also shows that there is no overfitting caused by the increase of the number of hyperparameters ($t$).

**Generalized adaptive diffusion convolution (GADC).** By now, we introduce the adaptive diffusion convolution based on heat kernel. Without loss of generality, we can have ADC extended to a generalized ADC (GADC), that is, not limiting the weight coefficients $\theta_k$ as heat kernel. Therefore, we have the feature propagation of GADC as:

$$\gamma^{(l)}(\hat{\mathbf{H}}, G)_i = \sum_{k=0}^{\infty} \theta_{ki}^{(l)} \mathbf{T}^k \hat{\mathbf{H}}_i \tag{13}$$

Table 1: Dataset Statistics [29]

|  | CORA | CiteSeer | PubMed | Coauthor CS | Amazon Computers | Amazon Photo |
|---|---|---|---|---|---|---|
| #Nodes | 2485 | 2110 | 19717 | 18333 | 13381 | 7487 |
| #Edges | 5069 | 3668 | 44324 | 81894 | 245778 | 119043 |
| #Classes | 7 | 6 | 3 | 15 | 10 | 8 |
| #Training-Nodes | 140 | 120 | 60 | 300 | 200 | 160 |
| #Validation-Nodes | 1360 | 1380 | 1440 | 4700 | 1300 | 1340 |
| #Test-Nodes | 985 | 610 | 18217 | 13333 | 11881 | 5987 |

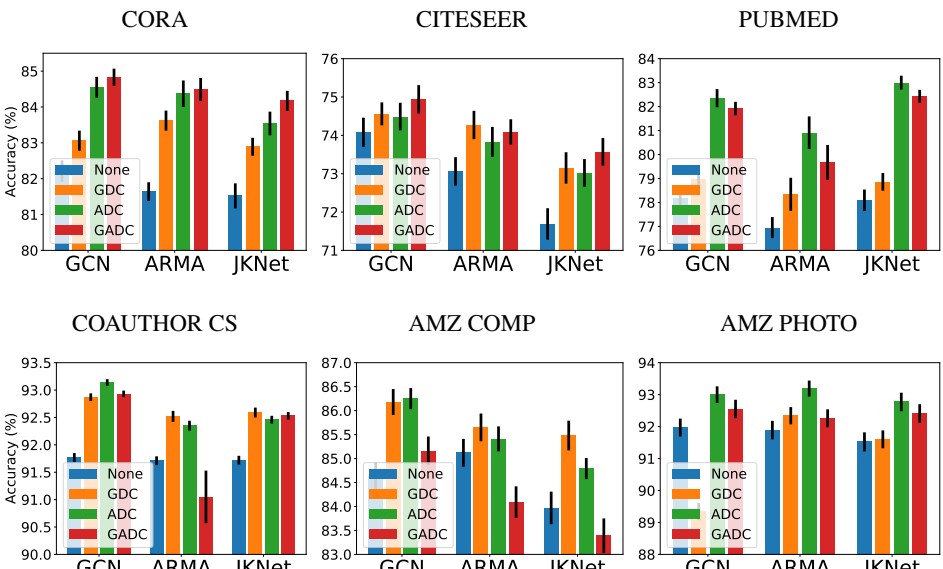

Figure 3: Semi-supervised node classification accuracy, on original model or improved with GDC or our trainable heat kernel. Our improvement surplus GDC in most datasets and models.

where $\theta_{ki}^{(l)}$ denotes the weight coefficient for $k$-hop neighbors on $l$-th layer and $i$-th channel. We restrict $\sum_{k=0}^{\infty} \theta_{ki}^{(l)} = 1$ during training.

**Implementation Details.** As we operate differently on each channel, whether we propagate before or after the feature transformation function actually matters. Empirically, we find that propagating on the input channels generates better results than propagating on the output channels (Cf. Figure 7 for details). Therefore, we swap the propagation and transformation steps in the original message passing networks from Eq. 1 to:

$$\mathbf{H}^{(l)} = \varphi^{(l)}(\gamma^{(l)}(\mathbf{H}^{(l-1)}, G)) \tag{14}$$

Additionally, calculating $e^{-\mathbf{L}t}$ directly is infeasible for large graphs. Practically, we need to use the top-$K$ truncation to approximate the heat kernel, making ADC (Eq. 12) and GADC (Eq.13) respectively updated as:

$$\gamma^{(l)}(\hat{\mathbf{H}}, G)_i = \sum_{k=0}^{K} e^{-t_i^{(l)}} \frac{\left(t_i^{(l)}\right)^k}{k!} \mathbf{T}^k \hat{\mathbf{H}}_i, \quad \gamma^{(l)}(\hat{\mathbf{H}}, G)_i = \sum_{k=0}^{K} \theta_{ki}^{(l)} \mathbf{T}^k \hat{\mathbf{H}}_i \tag{15}$$

Similar to GDC, ADC and GADC are flexible components that can be directly plugged into existing GNN models, enabling them to adaptively learn the neighborhood radius for feature aggregation.

# 4  Experiments

## 4.1  Experimental setup

We follow the standard procedure to conduct experiments and report results. Same as GDC, each result is averaged across 100 random data splits and initializations. All baseline GNNs use the same hyperparameters as GDC's baseline model. We only compare with the heat kernel version of GDC. We set the learning rate of $t$ equals to the learning rate of other parameters, which is 0.01. All results are calculated as averages with 95% confidence via bootstrapping.

The prediction task is focused on semi-supervised node classification. We use widely-adopted datasets including CORA, CiteSeer [28], PubMed [25], Coauthor CS, Amazon Computers and Amazon Photo [29]. Statistics of datasets are listed in Table 1. Same as GDC, we only use their largest connected components. The data is split to a development and test set. Development set contains 1500 nodes except Coauthor CS that contains 5000 nodes. The development set is split to a training set containing 20 nodes for each class and a validation set with remaining nodes.

We implement ADC and GADC on three models: GCN [19], JKNet [38] and ARMA [5]. We only replace original model's feature propagation function with ADC or GADC and preserve feature transformation function. The expansion step ($K$ in Eq.15) is set to 10. We use early stopping with patience of 100 epochs. Different from GDC, we don't use GAT [32] or GIN [37], because GAT uses learned attention matrix instead of adjacency matrix and GDC's experiments show that GIN performs much worse than other models.

## 4.2  Results

Figure 3 reports the main results when applying GDC and the proposed ADC on GCN, ARMA, and JKNet. It shows that ADC significantly improves base GNNs and outperforms GDC on most cases. Compared to GADC, which has $K\times$ more hyperparameters than ADC, ADC can match and achieve comparable or even better results.

In addition, we observe that the runs with low stopping epochs (less than 300) often perform worse than those with more stopping epochs. Forcing early stopping inactive in low epochs could help increase the performance by a little. This suggests that the neighborhood radius $t$ may need more training epochs, however, this contradicts with the early stopping strategy, which means high stopping epochs may cause $w$ to overfitting. To make a straightforward and fair comparison, we do not use this training trick in our experiments.

**Ablation study.**  The ablation studies on different diffusion setting are summarized in Figure 4. GDC can be seen as fixing $t$ heat kernel with sparsification. The results suggest that GDC's performance is partly due to its sparsification on propagation matrix, because if we remove sparsification, training $t$ for each feature channel and layer always performs better than fixing $t$ to its initialization value. Figure 4 also shows that by comparing the improvements brought by training $t$ at different levels, training $t$ for each channel and layer contributes the most to the performance improvements on the node classification task.

**The influence of rate between training and validation set.**  Because ADC trains a large number of hyperparameters on validation set, this may cause concerns about whether the improvement is due to overuse of the validation set. To verify that, we do the same experiment of Figure 3 on different rates between training and validation sets, with same settings. We fix the total number of nodes of the development set and change the number of nodes per class in training set. As stated before, the development set contains 1500 nodes except Coauthor CS which has 5000 nodes. Results in Figure 5 show that ADC constantly performs better than GDC and the original base GNN model (GCN) in most cases. This demonstrates that the advantage of ADC does not come from training more parameters on a large validation set.

**The influence of expansion step.**  Do the long-distance neighbor nodes really help? $K$ in Eq. 15 denotes the expansion step or truncation step of Taylor expansion. By changing $K$, we could examine whether neighbors further than $K$ step away really matter. Figure 6 shows that increasing the step of Taylor expansion ($K$) often helps to increase accuracy. Nodes further than ten step away don't carry valuable information.

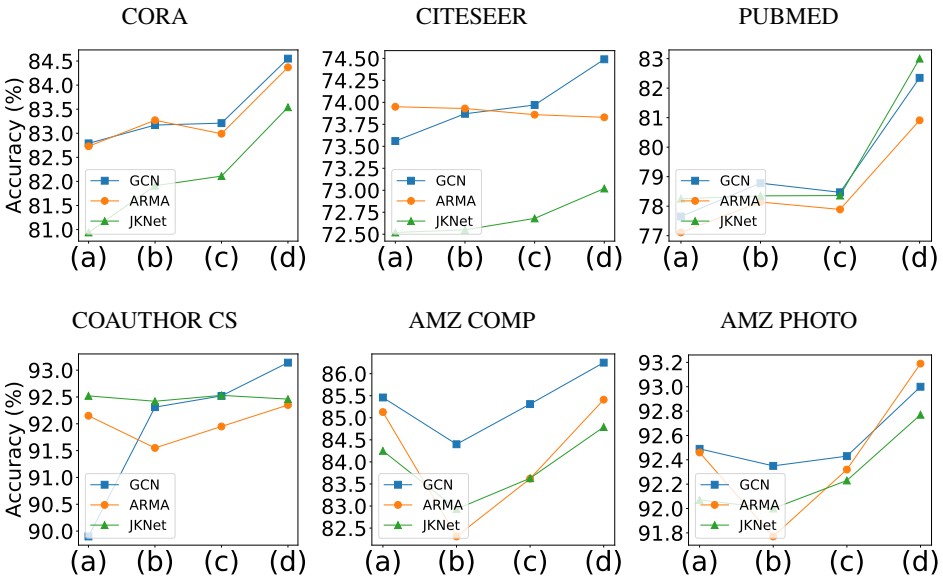

Figure 4: (a) Fixing $t$ to the initialization value. (b) Training one $t$ for all layers. (c) Training one $t$ for each layer. (d) Training a unique $t$ for each feature channel and layer. (a) can be seen as GDC without sparsification. Results show that after removing sparsification in GDC, training $t$ from data always helps improve performance.

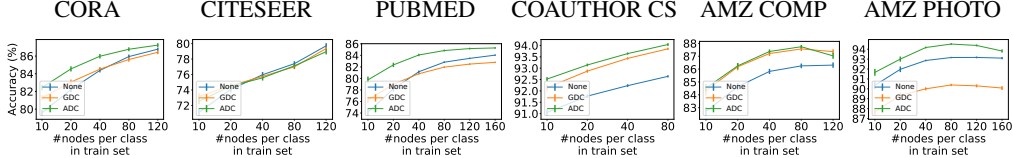

Figure 5: Influence of the number of training nodes. X-axis denotes the number of nodes per class in training set. We fix the total number of development set (train + valid) to be the same. ADC constantly performs better than GDC and the original model (GCN) in most cases. This indicates that the advantage of ADC is not due to training more parameters on a large validation set.

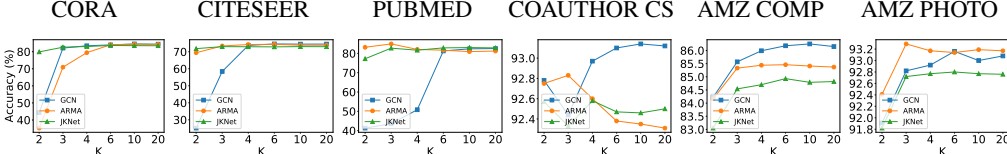

Figure 6: Influence of the number of Taylor expansion steps on the trainable heat kernel in ADC. Increasing step number helps increase the accuracy in most cases.

**The influence of propagation before or after transformation.** As we discuss in Eq.14, different from GCN, propagation before or after transformation does matter in our model, because this means learning unique $t$ for each input feature channel or each output feature channel. Figure 7 shows that propagation before transformation performs better than propagation after transformation. This is probably because input feature's channel carries more diverse information.

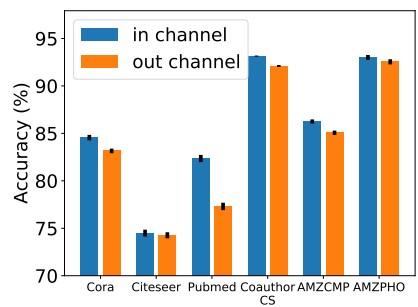

Figure 7: Comparing propagating before or after feature transformation function on ADC + GCN, which means learn a separate propagation $t$ for each input or output feature channel. Result shows that learning separate $t$ on input channels performs better.

# 5 Related Work

Bruna et al. [6] first propose a spectral graph convolutional network, using back propagation to learn the kernel filter. Later, ChebyNet [12] is proposed to leverage Chebyshev expansion to avoid the Laplacian eigendecomposition, which has high computational complexity. The graph convolutional network (GCN) [19] go one step further by simplifying the kernel filter to the first-order of Chebyshev expansion. The graph attention networks (GAT) [32] utilize self-attention layers to learn the importance of different nodes in neighborhood. GraphSAGE [15] generalizes graph convolution from transductive tasks to inductive tasks. Different types of feature propagation model have been proposed and used in practice [20, 13, 33, 7, 16].

In Chung et al.'s book [10], the properties of heat kernel in graph have been discussed thoroughly. David et al. [31] generalize windowed Fourier to graph with heat kernel. After GCN emerged, some graph convolution models integrating heat kernel have been proposed. GraphHeat [36] leverages heat kernel to enforce smoothness in the signal variation on graph, which acts as a low-pass filter. GDC [21] utilizes heat kernel as a special case of graph diffusion. In ProNE [39], Chebyshev expansion is proposed to approximate heat kernel.

Using learned coefficients of k-hops neighbors instead of hand-tuned has already been discussed in some other graph learning tasks. AdaDIF [4] proposes a class-specific adaptive diffusion method on label propagation. AGF [8], also in label propagation, uses adaptive graph filters to form a global decision by combining decisions from multiple graph filters. In node embedding tasks, an attention-based model [1] proposes to learn the length of random walk via backpropagation. PERDIF [26] learns a personalized diffusion over item models for top-n recommendation. AptRank [18] utilizes an adaptive diffusion method for protein function prediction. A recent work, GPR-GNN [9], utilizes adaptive generalized PageRank, but focusing on handle heterophily and over-smoothing. None of the previous methods calculates gradients on validation set to prevent overfitting or treat each feature channel and layer separately.

# 6 Conclusion and Discussion

In this work, we propose to learn the neighborhood radius for the feature aggregation process in GNNs. Traditionally, the neighborhood radius is either pre-determined, e.g., GCN, or hand-tuned, e.g., GDC. To make it practically applicable to real-world problem settings, we present a general GNN plugin ADC that can automatically learn the neighborhood radius for each GNN layer and each feature channel. Similar to GDC, ADC is able to enhance any graph-based model, particularly GNNs. By directly plugging ADC into existing GNNs, the experiments and ablation studies show that learning unique neighborhood radius for each feature channel in each GNN layer consistently and significantly improves the performance for downstream graph mining tasks.

Notwithstanding the promising results, future works might lie in improving the training procedure, as we notice that the neighborhood radius sometimes requires more epochs to train than other parameters. One possible direction is to study how to automatically balance this with early stopping strategy.

In terms of societal impacts, the proposed technique ADC shares the same promises and potential issues with the general GNN research. Graphs are naturally used for abstracting and modeling

relational and structured data, such as social networks, and graph neural networks, as a powerful tool for modeling graphs, may suffer from the privacy issues faced by the original social platforms or graph datasets. However, in general, we do not see ethical concerns or potential harms of the proposed technique.

For reproducibility, both the code and datasets (publicly available) used for experiments are included in the supplementary document.

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
