# OpenReview forum: "Adaptive Diffusion in Graph Neural Networks"
_NeurIPS.cc/2021/Conference — NeurIPS 2021 Poster_

### Official Review · Reviewer_uhXN · 2021-07-15

**Rating:** 8
**Confidence:** 4

**Summary:**

This paper proposes a strategy for adaptive diffusion convolution (ADC and GADC) that automatically learns the optimal neighborhood aggregation size, and it allows for learning a dedication propagate neighborhood for each GNN layer and each feature channel. They perform ablation studies and compares their method to related work on well-known benchmarks.

**Ethical Concerns:**

None.

**Limitations And Societal Impact:**

They address it adequately.

**Main Review:**

Originality: The ability to flexibly learn the neighborhood size is original and significant. This is an important contribution and plays very well into previous work as a natural and helpful extension.

Quality: The paper shows high quality through-out, it’s well done and the experiments are useful and convincing. I would appreciate some kind of visualization of the learned diffusion, that would be interesting. Also, it would be nice to see this on other benchmarks dealing with graph classification and edge classification.

Clarity: Clear writing and structure. The flow is nice and the material easy to digest.

Significance: I like this work and think it’s a significant contribution. There is however a lot of related work that is dealing with similar problems and I don’t know if there is something that they’ve missed citing and that’s very similar.

**Time Spent Reviewing:**

4

---

> ### Author Response · Authors · 2021-08-10
> **Response to Reviewer 4**
>
> Thanks for your recommendation for acceptance!
>
> **Reviewer Comment: I would appreciate some kind of visualization of the learned diffusion, that would be interesting.**
>
> Our Response: We will add a visualization of the distribution of neighborhood radius of each feature channel in the camera-ready version. Our first figure in Figure 1 is the change of average of neighborhood radius. This could be seen as a simple visualization of learning of diffusion. If you are interested in this figure, we could give a more detailed explanation. Similar to "learn t on train set", ADC's neighborhood radius still first decrease because other parameters are not well-learned at this time. The neighborhood radius has to shrink to make a lower loss. But as other parameters become more and more expressive, each node could now enlarge its receptive field to aggregate more information from distant neighbors.
>
> **Reviewer Comment: It would be nice to see this on other benchmarks dealing with graph classification and edge classification.**
>
> Our Response: Thank you for giving this suggestion. We are trying graph classification now and will provide the result as soon as possible.

---

### Official Review · Reviewer_is4Q · 2021-07-17

**Rating:** 7
**Confidence:** 3

**Summary:**

This paper proposes an adaptive diffusion convolution strategy that can automatically learn the optimal radius of message passing in graph neural networks and addresses the issue of selecting the optimal radius by hand-tuning. Additionally, this mechanism can serve as a plugin for graph neural networks and improve the model capacity and performance. Eventually, the paper validates this claim on the task of semi-supervised node classification.

**Limitations And Societal Impact:**

1.	There are two methods proposed in the paper, which are ADC and GDC. In the experimental session, comparisons are performed between them. Even though ADC is better than GDC in many cases, there are still some cases where GDC can excel. It would be great if the authors can offer some recommendations for these two methods.

**Main Review:**

1.	A novel mechanism to optimally select the neighborhood radius when performing graph diffusion convolution. The paper formulates choosing the radius as a bi-level optimization problem, and the optimization of the radius is performed on the validation set.
2.	It breaks the convention that graph convolution should be based on neighborhood and extends to freely learn an optimal radius for each layer and each feature channel.
3.	Thorough experiments are done to compare with the baseline GDC and explore the relationship between different radius updating mechanisms.


**Time Spent Reviewing:**

24

---

> ### Author Response · Authors · 2021-08-10
> **Response to Reviewer 3**
>
> Thanks for your positive feedback!
>
> **Reviewer Comment: It would be great if the authors can offer some recommendations for these two methods.**
>
> Our Response: The advantage of GDC is that it could work unsupervised models for unsupervised tasks like clustering. But ADC need back propagation to update neighborhood radius, so it has to be trained on supervised tasks.
>
> ADC could train on larger graph than GDC because ADC doesn't need to calculate the dense matrix (self-multiplication of adjacency matrix).

---

### Official Review · Reviewer_JPvA · 2021-07-17

**Rating:** 6
**Confidence:** 5

**Summary:**

This paper extends from graph diffusion convolution (GDC) that used hand-tuned neighborhood size for feature aggregation to adaptively learning the size. ADC can automatically learn the neighborhood radius for each GNN layer and
each feature dimension and acts as a plugin to any GNN model. Experimental results and ablation studies show
that learning a unique neighborhood radius for each feature channel in each GNN layer consistently
and improves the performance of downstream graph mining tasks.

**Limitations And Societal Impact:**

Yes.

**Main Review:**

Pros:
1. Improve from GDC that is manually tuned size for each graph by conducting grid search over the validation set, the paper proposes the adaptive diffusion convolution (ADC) to automatically learn the optimal neighborhood size from the data.
2. ADC chooses different neighborhood sizes for each GNN layer and feature channel.

Cons:
1. GDC applies diffusion on the original adjacent matrix to get a dense graph, and then sparsify the dense graph to get a sparse graph (Only one pass graph diffusion); while ADC extends such idea to each GNN layer over adjacent matrix. Therefore, the proposed method also has high computation complexity comparing to GDC. This is not mentioned in the paper.
2. The experimental setup does not follow standard splitting for semi-supervised node classification. Therefore it is difficult to compare the results in the paper to other recent STOA on the semi-supervised node classification (for example, "Direct Multi-hop Attention based Graph Neural Network", https://arxiv.org/pdf/2009.14332.pdf)
3. From the experimental results, the improvement seems not convincing and the reported accuracy improvement is GNN- and data-dependent: the original ARMA and JKNet are generally better than GCN on Cora, Citeseer. However, Fig. 3 did not show the benefit of JKNet (ARMA) vs GCN on Cora, Citeseer.  In fig. 3, the proposed method + ARMA (JKNet) is worse than (proposed method + GCN).  In comparison, GDC + ARMA (JKNet) is usually still better than GDC + GCN. Meanwhile, the proposed method + ARMA is also worse than the performance on COAUTHOR CS and AMZ COMP --- there are no explanations for these inconsistent results.
4. The performance numbers for ADC/GADC are not consistent, on some datasets, ADC is better, some GADC is better. Any explanation?
5. On page 5, feature transformation and propagation are swapped because "Empirically, we find that propagating on the input channels generates better results than propagating on the output channels". This conclusion might be data-dependent or GNN model-dependent: In figure 6,  the experiments were conducted using GCN. Is the same conclusion achieved on JKNet and ARMA models?

** I have read the responses from the authors and adjusted the score based on their answers to my questions.

**Time Spent Reviewing:**

6

---

> ### Author Response · Authors · 2021-08-10
> **Response to Reviewer 2**
>
> Thank you for your detailed and insightful comments! Please see our response below.
>
> **Reviewer Comment: GDC applies diffusion on the original adjacent matrix to get a dense graph, and then sparsify the dense graph to get a sparse graph (Only one pass graph diffusion); while ADC extends such idea to each GNN layer over adjacent matrix. Therefore, the proposed method also has high computation complexity comparing to GDC. This is not mentioned in the paper.**
>
> Our Response: ADC is actually faster than GDC because we don't need to calculate the dense matrix.
>
> The key difference between ADC and GDC is that we don't sparsify the matrix. As we stated in line 195, we use top-K truncation to approximate the heat kernel, which is equivalent to sparsify connections further than K step. Similar to Chebyshev expansion in ChebyNet and ProNE, this enables us to replace the time-consuming matrix multiplication with \textbf{multiplication between adjacency matrix and feature vector}. According to Eq. (15), we only need to calculate $\mathbf{T}^k \mathbf{H}$, where $\mathbf{T}$ is the normalized symmetric adjacency matrix and $\mathbf{H}$ is the hidden feature. This can be done by calculate $(\mathbf{T}...(\mathbf{T}(\mathbf{T} \mathbf{H})))$, which is just $k$ times the time complexity of $\mathbf{T} \mathbf{H}$. This also enables our model to work on larger graph than GDC.
>
> **Reviewer Comment: The experimental setup does not follow standard splitting for semi-supervised node classification. Therefore it is difficult to compare the results in the paper to other recent SOTA on the semi-supervised node classification (for example, "Direct Multi-hop Attention based Graph Neural Network", https://arxiv.org/pdf/2009.14332.pdf).**
>
> Our Response: Many recent studies have shown that multiple random splittings is better than standard splitting.
>
> "Pitfalls of Graph Neural Network Evaluation" finds that standard splitting favors the model that overfits the most. APPNP finds that a lot of works have suffered from superficial statistical evaluation and experimental bias from overfitting, which is caused by using single training/validation/test split. By calculating averages with 95% confidence via bootstrapping on 100 random splits, we think we made a fair comparison between origin model, GDC and ADC.
>
> Furthermore, we are not focusing on getting the highest SOTA performance. Our contribution is to show that learning neighborhood sizes for each GNN layer and feature channel could really benefit.
>
> **Reviewer Comment: From the experimental results, the improvement seems not convincing and the reported accuracy improvement is GNN- and data-dependent: the original ARMA and JKNet are generally better than GCN on Cora, Citeseer. However, Fig. 3 did not show the benefit of JKNet (ARMA) vs GCN on Cora, Citeseer. In fig. 3, the proposed method + ARMA (JKNet) is worse than (proposed method + GCN). In comparison, GDC + ARMA (JKNet) is usually still better than GDC + GCN. Meanwhile, the proposed method + ARMA is also worse than the performance on COAUTHOR CS and AMZ COMP --- there are no explanations for these inconsistent results.**
>
> Our Response: Thanks for pointing this out. Our proposed method doesn't outperform GDC in all datasets with all models. There are two reasons.
>
> The first reason is that we remove the sparsification in GDC to make the propagation matrix learnable. In our ablation study (Figure 4), column (a) fixing t setting is equivalent to GDC without sparsification. We show that after removing sparsification, adaption consistently helps improve the performance.
>
> Another reason may lies in the difficulty to learn neighborhood radius. As we state in line 223 to 228, we observe that neighborhood radius need more epochs to learn than other parameters. However, training too much epochs will cause other parameters to overfit on train set, and this is why GCN needs early stop. One brute force solution we have tested is that each time ADC reach early stop, we random split the train and valid set again and reset all other parameters to random initialization, but preserve the learned neighborhood radius and keep updating neighborhood radius on new setting. By learning neighborhood radius on multiple random split and initialization of other parameters, ADC could greatly improve the performance. However, we don't use this kind of trick in evaluation part to make a fair comparison.
>
> **Reviewer Comment: The performance numbers for ADC/GADC are not consistent, on some datasets, ADC is better, some GADC is better. Any explanation?**
>
> Our Response: We think this shares similar explanation with last question. As GADC has $K \times$ more neighborhood-radius parameters than ADC, GADC needs more time to converge to optimal solution, however, this is contradict to early stop strategy. This explains why GADC performs well on small datasets like Cora and Citeseer but fails on large graph.
>
> **Reviewer Comment: On page 5, feature transformation and propagation are swapped because "Empirically, we find that propagating on the input channels generates better results than propagating on the output channels". This conclusion might be data-dependent or GNN model-dependent: In figure 6, the experiments were conducted using GCN. Is the same conclusion achieved on JKNet and ARMA models?**
>
> Our Response: We are trying this now and will report it as soon as possible!

---

> > ### Author Response · Authors · 2021-08-15
> > **Additional Result**
> >
> > Here are the updated results for difference of propagating on the input channels or output channels on JKNet and ARMA:
> >
> > |               | CORA              | CITESEER          | PUBMED            | CoauthorCS        | AMZCMP            | AMZPHO            |
> > | ---           | ---               | ---               | ---               | ---               | ---               | ---               |
> > | JKNET(input)  | **83.54 +- 0.33%**    | **73.02 +- 0.36%**    | **83.00 +- 0.29%**    | 92.46 +- 0.07%    | **84.79 +- 0.22%**    | **92.77 +- 0.29%**    |
> > | JKNET(output) | 81.95 +- 0.31%    | 72.66 +- 0.43%    | 79.48 +- 0.34%    | **92.56 +- 0.10%**    | 84.02 +- 0.26%    | 92.46 +- 0.31%    |
> > | ARMA(input)   | **84.37 +- 0.37%**    | 73.83 +- 0.39%    | **80.91 +- 0.68%**    | **92.35 +- 0.09%**    | **85.41 +- 0.26%**    | **93.19 +- 0.25%**    |
> > | ARMA(output)  | 83.34 +- 0.25%    | **74.00 +- 0.36%**    | 78.35 +- 0.38%    | 92.35 +- 0.08%    | 85.04 +- 0.29%    | 92.70 +- 0.28%    |
> >
> > We could find that propagating on the input channels performs better in most cases.

---

### Official Review · Reviewer_R6WW · 2021-07-18

**Rating:** 6
**Confidence:** 4

**Summary:**

This paper introduces adaptive graph diffusion convolution (ADC), a new module that adaptively and automatically determines the neighborhood size for the better feature aggregation of existing GNN frameworks. It extends the graph diffusion convolution (GDC) one step further to make the selection of neighborhood radius learnable. Generally speaking, it is an interesting idea to directly learn different neighborhood radius in different layers. This paper is well-written and easy to follow, and related works are covered comprehensively. Authors further conduct (limited) experiments comparing with selected baselines in node classification tasks, various analysis, and ablation studies demonstrate the strength of this proposed module.

**Limitations And Societal Impact:**

There is no obvious negative societal impact found.

**Main Review:**

My main concerns of this work lie in two aspects. First, the technical novelty is rather limited since the backbone of the model basically follows its predecessor – GDC, including the design graph diffusion, the concept of neighborhood radius, and the heat kernel. It seems like the contribution of this work is merely the proposal of an adaptive version of choosing the neighborhood radius. Second, the authors claim that ADC can be utilized as a plug-in for all GNN layers. However, based on the design of ADC, it leverages the weighted self-multiplication of the Laplacian matrix to achieve the idea of graph diffusion, which makes it hard to apply in non-Laplacian matrix-based GNNs (e.g., GAT and GTN) (as also mentioned by authors in Section 4.1).

There are some additional comments:
1.	Eq. (6) confuses me. It is not obvious how t equals the neighborhood radius for the heat kernel-based GDC. I double-checked the GDC paper and found no evidence of such a claim. Either a detailed explanation or a reference can be provided here.
2.	Since one of the claims in this work is the development of a novel GNN plug-in, it is not adequate of using only one task to demonstrate its ability. Other than node classification, experiments on different tasks (node clustering, link prediction) are expected.
3.	As mentioned in Related Work, the graph diffusion convolution shares lots of commonalities with the conventional graph diffusion method on label propagation. Authors should also consider them [4], [8], [9] or demonstrate how they cannot be compared in this task.
4.	Legends in Fig. 5 are not consistent with the context.


**Time Spent Reviewing:**

3 hours

---

> ### Author Response · Authors · 2021-08-10
> **Response to Reviewer 1**
>
> Thank you for the detailed comments. We provide detailed answers to each question as follows:
>
> **Reviewer Comment: First, the technical novelty is rather limited since the backbone of the model basically follows its predecessor – GDC, including the design graph diffusion, the concept of neighborhood radius, and the heat kernel. It seems like the contribution of this work is merely the proposal of an adaptive version of choosing the neighborhood radius.**
>
> Our Response: The misunderstanding is that the concept of neighborhood radius is first proposed by us in this paper. It can be served as a metric to measure how far feature can propagte in each layer, for both diffusion and multi-hop network. This can be very helpful for future research on graph propagation.
>
> We think making choosing the neighborhood radius adaptively is both significant and non-trivial.
>
> Significant: We are the first work which personalizes propagation methods for each feature channel and layer. This can be more important for industrial datasets, in which node feature may be a concatanetion of features from different sources. This makes one neighborhood radius impossible to fit all feature channels.
>
> Non-trivial: Multiple techniques have been introduced by us to make adaptive neighborhood radius usable. First, different from GDC's calculating heat kernel directly, we use top-K truncation of heat kernel (Eq. 15) to make the gradient of neighborhood radius calculable. Then, we separate datasets for neighborhood radius and other parameters to prevent overfit. Finally, we propose a two step gradient descent (Eq. 10, 11) to approximate the time-consuming bilevel optimization problem.
>
> **Reviewer Comment: Second, the authors claim that ADC can be utilized as a plug-in for all GNN layers. However, based on the design of ADC, it leverages the weighted self-multiplication of the Laplacian matrix to achieve the idea of graph diffusion, which makes it hard to apply in non-Laplacian matrix-based GNNs (e.g., GAT and GTN) (as also mentioned by authors in Section 4.1).**
>
> Our Response: ADC could also work on non-Laplacian matrix-based GNNs. GDC's authors make evaluation on GAT by multipling GDC's edge weight with the learned attention weight. But we think this method isn't straightforward and could be seemed as an architecture change, so we didn't include it in evaluation part. But we are trying this now and will report it in the discussion as soon as we get result.
>
> **Reviewer Comment: Eq. (6) confuses me. It is not obvious how t equals the neighborhood radius for the heat kernel-based GDC. I double-checked the GDC paper and found no evidence of such a claim. Either a detailed explanation or a reference can be provided here.**
>
> Our Response: The definition of neighborhood radius is Eq. (2). In line 101, we state that for heat kernel, $\theta_k = e^{-t}\frac{t^k}{k!}$. Putting this into the definition of neighborhood radius, we could get the neightborhood radius $r_h$ of heat kernel version of GDC is:
>
> $$r_h = \frac{\sum_{k=0}^\infty e^{-t}\frac{t^k}{k!} k}{\sum_{k=0}^\infty e^{-t}\frac{t^k}{k!}} = \frac{e^{-t}\sum_{k=0}^\infty \frac{t^k}{k!} k}{e^{-t}\sum_{k=0}^\infty \frac{t^k}{k!}} =  \frac{e^{-t} (e^t t)}{e^{-t} e^t} = t$$
>
> Maybe it's better if we use a different sign to distinguish neighborhood radius of heat kernel $r_h$ and general neighorhood radius $r$ in our paper. We will change this in the camera-ready version.
>
> **Reviewer Comment: Since one of the claims in this work is the development of a novel GNN plug-in, it is not adequate of using only one task to demonstrate its ability. Other than node classification, experiments on different tasks (node clustering, link prediction) are expected.**
>
> Our Response: Our model need supervised tasks to learn the adaptive neighborhood radius. We are testing on graph classification tasks and will report it as soon as possible.
>
> **Reviewer Comment: As mentioned in Related Work, the graph diffusion convolution shares lots of commonalities with the conventional graph diffusion method on label propagation. Authors should also consider them [4], [8], [9] or demonstrate how they cannot be compared in this task.**
>
> Our Response: We think none of these works calculate gradients on validation set to prevent overfitting or treat each feature channel and layer separately. These two contributions are key factors that make adaptive diffusion useful. Without training on validation set, models need to use hand-tuned regularization on neighborhood radius for every dataset to prevent overfit (as stated in AdaDIF and GDC). And in our ablation study part, we show that training different neighborhood radius in different layers and in different channels contributes the most to the performance improvement.
>
> **Reviewer Comment: Legends in Fig. 5 are not consistent with the context.**
>
> Our Response: Thank you. We have fixed that in our updated version of paper.

---

> > ### Author Response · Authors · 2021-08-24
> > **Additional Result**
> >
> > Here are the updated results for ADC's improvement on GAT:
> >
> > |               | CORA              | CITESEER          | PUBMED            | CoauthorCS        | AMZCMP            | AMZPHO            |
> > | ---           | ---               | ---               | ---               | ---               | ---               | ---               |
> > | GAT  | 82.66 +- 0.28%    | 72.88 +- 0.38%    | 78.03 +- 0.31%    | 69.57 +- 3.77%    | 60.98 +- 4.89%    |  64.61 +- 6.06%   |
> > | GAT+ADC | 83.29 +- 0.28%    | 73.35 +- 0.31%    | 78.72 +- 0.29%    | 88.50 +- 0.11%    | 41.53 +- 1.94%   | 56.16 +- 5.08%    |
> >
> > We find that GAT breaks down on CoauthorCS, AMZCMP and AMZPHO. Similar phenomenons also have been reported in graph diffusion convolution. ADC could help enhance GAT except AMZCMP and AMZPHO, which may caused by very early stopping epoch, leaving no time for ADC to learn neighborhood radius.

---

> > > ### Comment · Reviewer_R6WW · 2021-08-31
> > > **Feedback to the new results**
> > >
> > > Thanks to the authors for the newly added results. Together with the answers to my other concerns, they essentially addressed the issues I raised up so I would love to improve the score by 2 points.

---

### Decision · Program_Chairs · 2021-09-27

**Decision:**

Accept (Poster)

**Comment:**

This paper proposes a method called adaptive graph diffusion convolution (ADC), which can adaptively decide the neighborhood size for each layer and feature channel. The proposed method can be plugged in to any GNN frameworks. The experimental studies have demonstrated the superiority of the proposed module. The reviewers were split in the beginning, with concerns about novelty etc. After receiving a strong rebuttal from authors, including additional experiments, two reviewers have raised their scores to reflect their satisfaction to the rebuttal. In the end, the reviewers have reached consensus to accept this paper.